# Quantum local random networks and the statistical robustness of quantum scars

Federica Maria Surace[1,2*], Marcello Dalmonte[1,2], Alessandro Silva[1]

**1** International School for Advanced Studies (SISSA), via Bonomea 265, 34136 Trieste, Italy
**2** The Abdus Salam International Centre for Theoretical Physics (ICTP), strada Costiera 11, 34151 Trieste, Italy
* fsurace@caltech.edu

December 19, 2022

## Abstract

We investigate the emergence of quantum scars in a general ensemble of random Hamiltonians (of which the PXP is a particular realization), that we refer to as quantum local random networks. We find a class of scars, that we call "statistical", and we identify specific signatures of the localized nature of these eigenstates by analyzing a combination of indicators of quantum ergodicity and properties related to the network structure of the model. Within this parallelism, we associate the emergence of statistical scars to the presence of "motifs" in the network, that reflects how these are associated to links with anomalously small connectivity. Most remarkably, statistical scars appear at well-defined values of energy, predicted solely on the base of network theory. We study the scaling of the number of statistical scars with system size: by continuously changing the connectivity of the system we find that there is a transition from a regime where the constraints are too weak for scars to exist for large systems to a regime where constraints are stronger and the number of statistical scars increases with system size. This allows to define the concept of "statistical robustness" of quantum scars.

# 1   Introduction

Recently a great deal of research has focused on the fundamental concepts of thermalization and ergodicity shifting the focus from many-body spectra [1, 2] to the dynamics of observables [3]. A cornerstone of this program has been the formulation of the eigenstate thermalization hypothesis (ETH) [4, 5], which identifies the statistical properties of matrix elements of observables with the observation of thermal behaviour in their expectation values and correlation functions. More recently, the conditions of quantum chaos in many body systems have been further refined with the introduction of out-of-time-order correlations (OTOC) [6] and adiabatic gauge potentials [6, 7].

While two broad classes of systems have been introduced, nonergodic/localized [8, 9] vs. thermalizing [3], several systems have been shown to display intermediate behavior, where ETH is satisfied only at sufficiently high energies or for portions of the spectrum (weak ETH [10]). A prominent example in this class are quantum scars [11], that are non-ergodic eingestates embedded within the ergodic continuum. While those states are irrelevant for thermodynamics, they can still lead to very specific, intrinsically many-body phenomena in quantum quench experiments, provided the initial state has a significant overlap with them [12]. These states, which are qualitatively similar to localized states rarely occurring in the delocalized continuum of Anderson-type models [13], have been predicted in a variety of specific models [14–22], starting with constrained ones such as the PXP model [23–33].

As for localized states in the delocalized continuum, it was recently found that quantum scars of the PXP model are unstable against perturbations, suggesting that their occurrence might need fine tuning [34, 35]. It is thus presently unclear whether scarring is a robust phenomenon (and if so, in which sense), or if it generically requires parameter tuning to survive the thermodynamic limit.

While the approach to quantum scarring typically pivots around the analysis of spectral properties of 'deterministic' models, here, we pursue a different approach, and analyze the robustness of scar manifolds statistically. It was already noticed that the analysis of the network representation of the Hamiltonian is particularly convenient to understand many properties of constrained models displaying scars [23] (or even shattering of the Hilbert space [36, 37]) :

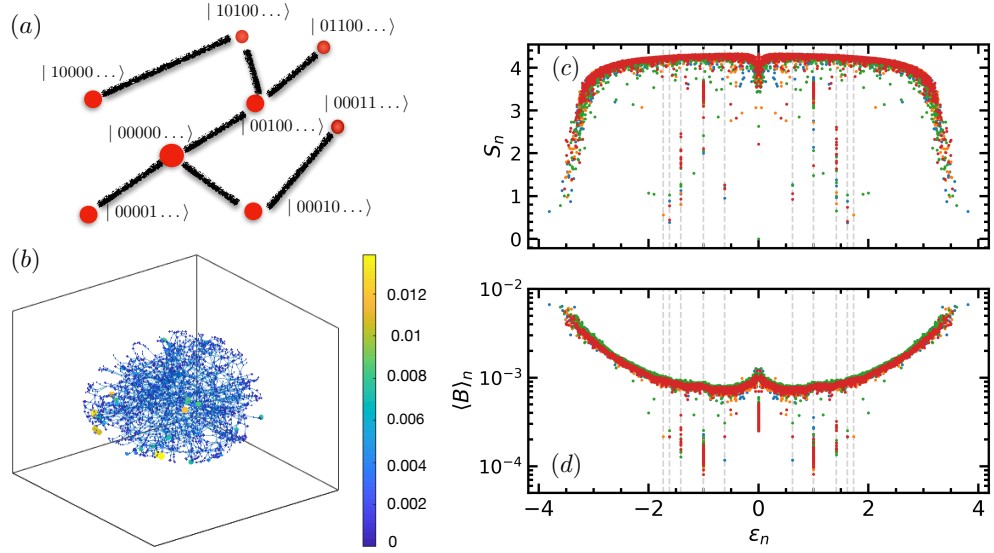

Figure 1: (a) Illustrative example of a QLRN: states differing by a single spin flip have probability $p$ of being connected by an edge. (b) Graphical representation of an eigenstate with $\epsilon = 0$ localized in the periphery of the network for $N = 13$, $p = 0.15$. The color indicates the weight of the eigenstate on each node. The weight is concentrated on few nodes that are loosely connected with the rest of the network. (c) Bipartite entanglement entropy and (d) betweenness centrality of the eigenstates vs their energy for $p = 0.15$, $N = 14$. The different colors refer to different realizations. Dashed grey lines indicate the special energies ($\epsilon^* = \pm 1, \pm\sqrt{2}, \pm(\sqrt{5} \pm 1)/2, \pm\sqrt{3}$) associated with statistical scars. At these energies, degenerate eigenstates are found, whose entanglement entropy and betweenness centrality are anomalously small with respect to the other eigenstates belonging to the thermal cloud.

these models are geometrically equivalent to networks with a number of nodes exponentially large in system size $N$, but an average degree per node only linear in $N$ as a result of the locality of the Hamiltonian [24]. We build upon this analogy to define a general ensemble of Hamiltonians, called *quantum local random network models*, which includes the PXP model as a particular realization. Hamiltonians belonging to this ensemble are the adjacency matrices of networks whose nodes are indexed by a string of quantum numbers (e.g., $\{01001\ldots\}$) while edges are drawn randomly with probability $p$ only among vertices differing by local moves (spin flips): in this way, the constraints are statistically encoded in the dynamics. The probability $p$ represents a continuous parameter that quantifies the strength of the constraints (a small $p$ indicates strong constraints, and vice versa).

We study in detail the spectra and the corresponding eigenfunctions and prove that generic Hamiltonians in this class can display *statistical* scars, a class of eigenstates that are localized on the network. Statistical scars occur always at specific energies $\epsilon^* = 0, \pm 1, \pm\sqrt{2}, \pm\sqrt{3}, \pm(\sqrt{5} \pm 1)/2, \ldots$, whose values are governed by spectral graph theory [38, 39]. A study of the scaling of the average degeneracy of statistical scars as a function of system size shows the occurrence of a series of eigenstate phase transitions as a function of $p$ between phases in which scars at a given energy $\epsilon^*$ proliferate and phases in which their number decreases.

## 2    Quantum many-body scars

The first model in which quantum scars were discovered is the PXP model [23] which, on a chain of $N$ sites with open boundary conditions, is defined by

$$H_{PXP} = \sum_{i=1}^{N-2} P_i X_{i+1} P_{i+2} + X_1 P_2 + P_{N-1} X_N, \tag{1}$$

where $P_i = (1 - Z_i)/2$ and $X_i, Z_i$ are local Pauli matrices. The dynamics of the PXP model is highly constrained (reflecting the microscopic mechanism of Rydberg blockade [12, 40]): it is impossible to flip a spin from down to up, if one of its nearest neighbours is up. Interestingly, the model in the subspace containing the spin-down state $|\circ \circ \circ \dots\rangle$ can be represented as a tight-binding Hamiltonian on a specific network (Fibonacci or Lucas cube) [24].

While the majority of scars, identified through their overlap with the $\mathbb{Z}_2$ state $|\circ \bullet \circ \bullet \dots\rangle$ and their low entanglement entropy, feature size-dependent effects, it was recently shown [29] that this model possesses also a few exact scar states (of the form of exact matrix product states) in the thermodynamic limit at the special energies $\epsilon = 0, \pm\sqrt{2}$. Individual scars are unstable with respect to perturbations: perturbations respecting the symmetries of the PXP model make them evaporate in the continuum of ergodic states. The instability of individual scars does not imply however that deformations of the PXP model cannot possess a *scar manifold*, i.e., a set of non-ergodic, low-entangled states immersed in the ergodic continuum, which are *not* continuous deformations of PXP scars. In this case, the existence of a scar manifold as a whole could be described as *statistically robust*.

## 3    Quantum local random networks

In order to address the question of statistical robustness of quantum scars, we notice that a common tract of constrained models is their representability as hopping Hamiltonians on networks whose nodes are indexed in the computational basis ($|\{\sigma\}\rangle$ with $\sigma = \circ, \bullet$ for the PXP model). It is therefore appealing to embed the PXP in a much broader ensemble of Hamiltonians, which we call *Quantum Local Random Networks* (QLRN) sharing the common ingredients of *locality* (in a way we specify below) and *constrained dynamics*.

Let us illustrate the construction of a QLRN in the simplest case (see Fig. 1-(a)): consider the network whose vertices are the sequences of $N$ elements $\{\sigma_i\}$, where $\sigma_i = 0, 1$ and $i = 1, \dots, N$, representing the computational basis of the Hilbert space of a spin system. Each pair of vertices is connected by an edge with probability $0 \le p \le 1$ provided they differ by a single flip of a boolean variable. The Hamiltonian for a model of this type reads

$$H = \sum_i X_i \sum_{\{\sigma\}} s_i^{\{\sigma\}} |\{\sigma\}\rangle \langle\{\sigma\}| , \tag{2}$$

where $s_i^{\{\sigma\}}$ are random variables that can assume the values 0 or 1 with probability $1 - p$ and $p$ respectively. These variables satisfy $s_i^{\{\sigma\}} = s_i^{\{\sigma'\}}$ for $|\{\sigma'\}\rangle = X_i |\{\sigma\}\rangle$, such that the Hamiltonian is Hermitian, but are otherwise independently distributed.

The adjacency matrix of the resulting network is then the Hamiltonian whose spectrum and eigenfunctions will be the subject of our study. We note that, in this context, locality is

intended in the sense that states connected by the Hamiltonian only differ by the properties of a single site $i$, but in general the Hamiltonian does not have a representation as a sum of terms with finite support. Evidently, the PXP model is a particular realization of a QLRN with $p = 0.25$, since in this model only one in four configurations of the nearest neighbours of a spin allows it to be flipped.

Note that, similarly to the PXP model, each Hamiltonian of the QLRN ensemble has matrix elements only between states with opposite $Z$ parity, and hence anticommutes with the operator $\mathcal{C} = \prod_i Z_i$. As a consequence, the spectrum is symmetric around $\epsilon = 0$. Another consequence is that, from the point of view of network theory, in a QLRN the clustering coefficient of each node (which is proportional to the number of triangles through the node [41]) is always zero, because the nearest neighbours of a vertex have the same parity, so they cannot be joined by an edge.

The construction of a QLRN can also be generalized to larger local Hilbert space dimensions (see Appendix A). We leave the study of these generalized QLRNs to future works.

# 4 Localized eigenstates

The use of the language of network theory in condensed matter physics has a long history, starting from studies of Anderson-type localization in generic networks [42], disorder-free localization on random trees [38] or as a function of clustering coefficient [43,44]. The possibility to generate localized states without disorder by taking advantage of geometrical constraints suggests that models of this type could be of interest for numerous problems, as was recently recognized in the context of the physics of many-body localization [45] and thermalization [46,47].

To study how the physics of localization emerges in a QLRN, we analyze the spectrum numerically for a finite size $N$ at different $p$. We first consider the density of states (DOS) (see Appendix B): while for $p = 1$ the spectrum is obviously the sequence of peaks associated to a spin of size $N$ in unit magnetic field, as $p$ diminishes the peaks first broaden, merging in a bell shaped DOS with a clear delta-function peak at $\epsilon = 0$. This peak was also observed in the DOS of tight-binding models defined on random Erdös-Rényi networks [43], where it was associated with localized states. We remark, however, that localization is not the only possible origin of this delta peak, and we will have to consider other quantities (such as the participation ratio) to prove the emergence of localized eigenstates. Another mechanism that may lead to the presence of a large degeneracy at $\epsilon = 0$ is, for example, the interwining of spectral reflection symmetry [48] and an ordinary symmetry of the Hamiltonian: while QLRNs have spectral reflection symmetry, in general they do not possess other symmetries (e.g., inversion), and therefore we cannot use this property to argue that their zero-energy eigenspaces have to be exponentially degenerate.

In the case of QLRN one has to pay attention to a trivial type of localization associated to disconnected vertices which get isolated as $p$ diminishes (a phenomenon similar to the fragmentation of Hilbert spaces observed in Ref. [36,49], as shown in Appendix C). Since in this work we will be interested in non-trivial localized states on QLRN and their connection to the physics of scars, in the following we will always identify the giant connected component of a QLRN and study localized states in this subspace. As expected a peak at $\epsilon = 0$ in its spectrum is present also under this restriction. We find that within this degenerate subspace it

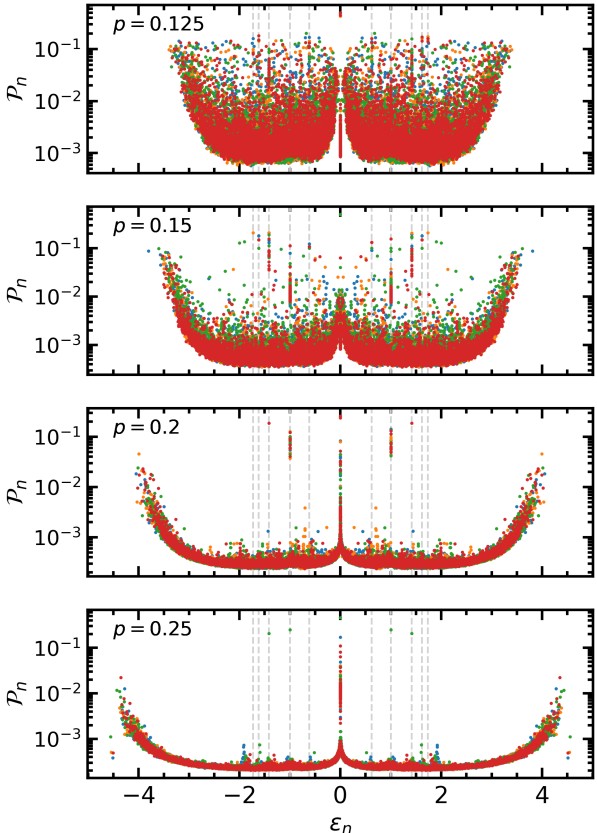

Figure 2: Participation ratio $\mathcal{P}_n$ of the eigenstates for different values of $p$ for system size $N = 14$. The colors indicate different realizations of the network. Statistical scars have large value of $\mathcal{P}_n$: they are localized in the computational basis. For large $p$ their number goes to zero.

is possible to find non-trivial eigenstates localized on the *periphery* of the network as depicted in Fig. 1-(b).

The localized states at $\epsilon = 0$ are just the simplest of a class of nontrivial localized states on the QLRN emerging at sufficiently small $p$. In order to characterize the localization properties of these and other eigenstates one may write them in the computational basis $|\Psi_n\rangle = \sum_{\{\sigma\}} c_n(\{\sigma\})|\{\sigma\}\rangle$ and study the participation ratio

$$\mathcal{P}_n = \sum_{\{\sigma\}} |c_n(\{\sigma\})|^4. \tag{3}$$

In addition, the structure of wave functions on the QLRN can be studied using standard measures of the character of nodes: *i)* the degree $k(\{\sigma\})$, i.e., the number of connections that the node $\{\sigma\}$ has to other nodes; *ii)* the centrality $C(\{\sigma\}) = 1/\sum_{\sigma' \neq \sigma} l_{\sigma'\sigma}$, where $l_{\sigma'\sigma}$ is the distance between two nodes in the network, which characterizes how close a node $\{\sigma\}$ is to the other nodes; and *iii)* the betweenness centrality $B(\{\sigma\})$ defined as the number of shortest paths among different vertices passing through $\{\sigma\}$, which quantifies how "central" a given node is in the network. One can easily use these quantities to study eigenstates by defining

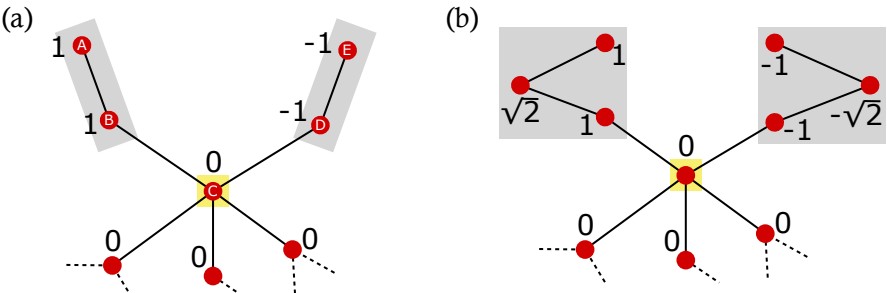

Figure 3: Graphical representation of examples of localized eigenstates with energy (a) $\epsilon^* = 1$ and (b) $\epsilon^* = \sqrt{2}$.

their averages over a generic eigenstate $|\Psi_n\rangle$, e.g., for the betweenness

$$\langle B \rangle_n = \sum_{\{\sigma\}} | c_n(\{\sigma\}) |^2 B(\{\sigma\}). \tag{4}$$

Finally, since these eigenstates can be interpreted as many-body states of a spin system of size $N$ one may compute the half- system entanglement entropy $\mathcal{S}_n$ to connect localization on QLRN to the physics of scars.

As seen in Fig. 1-(c) by plotting the half-chain entanglement entropy $\mathcal{S}_n$ for a QLRN at $p = 0.15$ as a function of eigenstate energy $\epsilon$ one can easily identify a number of eigenstates whose $\mathcal{S}_n$ is significantly lower than the typical value at that energy, therefore behaving as quantum scars. Most of these eigenstates share the feature of having significant (and untypical) participation ratio (see Fig. 2), and are therefore localized on the network. The participation ratio of the other eigenstates decreases with system size and gets closer to a thermal cloud with a smooth dependence on $\epsilon$ (see Appendix D). The untypical localized eigenstates have another remarkable property, as shown by plotting the eigenstate average betweenness $\langle B \rangle_n$ vs. $\epsilon$ (Fig. 1-(d)). Localized scars at specific energies (vertical lines at $\epsilon^\star = 0, \pm 1, \pm\sqrt{2}, \pm(\sqrt{5} \pm 1)/2, \pm\sqrt{3}, \dots$) tend to have a lower betweenness than the rest, indicating that they are not just localized, but localized on the periphery of the network: those are the key features that define statistical scars. Similar features are observed in the degree and closeness centrality of the eigenstates (see Appendix E). As shown in Fig. 2, statistical scars proliferate as $p$ is lowered below a certain threshold $p \simeq 0.2$.

## 5 Statistical scars

We now further investigate the presence of statistical scars at specific energies. The special energies $\epsilon^\star$ are well known to be the eigenvalues of the adjacency matrices of small trees [38,39]. The fact that various figures of merit, including the centrality and degree (see Appendix E), suggest that statistical scars are localized on the periphery of the network, indicates that small elementary subgraphs (motifs) might be the basic elements associated to statistical scars. This is indeed the case as shown in Fig. 3: the number next to each node is the coefficient $c_n(\{\sigma\})$ of the eigenstate $|\Psi_n\rangle$ in the computational basis. Each state is localized in the grey rectangles: all other nodes have $c_n(\{\sigma\}) = 0$. In both examples, the eigenstate of the full graph is constructed using as building block an eigenstate of a small motif (of two sites

in (a) and three sites in (b)): the graph contains two copies of the motif; the coefficient of the motif eigenstate are assigned with opposite signs on the two copies. More general networks with the same eigenstates can be constructed by adding edges to the graphs depicted here, provided that, for each node not belonging to the grey subgraph, the sum of the coefficients of its neighbours is 0. These examples show that the eigenfunction of subgraphs of two vertices (eigenvalues $\pm 1$) or three vertices (eigenvalues $\pm\sqrt{2}, 0$) can be easily incorporated into eigenfunctions of the whole QLRN whenever geometrical structures of the type of Fig. 3-a or Fig. 3-b occur on its periphery. The construction can be generalized to all the energies $\epsilon^*$ that are eigenvalues of small motifs (see Appendix F). These motifs are responsible for the presence of statistical scars: this claim is corroborated by the numerical observation that the degeneracy of statistical scars almost coincides with the number of occurrences of the (duplicated) motifs for accessible system sizes [1] (see Fig. 4 and Appendix F).

The occurrence of network motifs associated to statistical scars depends both on the overall system size $N$ and, most crucially, on $p$. In order to investigate how many scars are to be expected as a function of system size, we studied how the degeneracy of statistical scars ($\mathcal{N}_{\mathrm{scars}}$) with a given value of $\epsilon^\star$ and the number of occurrences of the associated motifs ($\mathcal{N}_{\mathrm{motifs}}$), averaged over realizations of the QLRN, scale with $N$ for a fixed $p$. This is shown in Fig. 4 for $\epsilon^\star = 1$: while for $p = 0.25$ both $\mathcal{N}_{\mathrm{motifs}}$ and $\mathcal{N}_{\mathrm{scars}}$ decrease with $N$, a completely different behavior, characterized by a continuous growth, is seen for smaller $p$. This fact seems to suggest the presence of an eigenstate transition as a function of $p$. The transition is the result of the competition between two effects: when the system size grows, motifs are less dense (the probability of having $m$ nodes disconnected from the rest of the network decreases as $(1-p)^{mN}$) but the number of nodes in the network grows ($\sim 2^N$). With this argument, we can estimate the number of motifs (see Appendix F)

$$\mathcal{N}_{th}(\epsilon^* = 1) = 2^N (1-p)^{4N-6} p^4 \frac{N(N-1)}{2}(N-1)^2. \tag{5}$$

We hence obtain the transition point $p_c(\epsilon^* = 1) = 1 - 2^{-1/4} \simeq 0.1591$, in agreement with the numerical data of Fig. 4. A similar behaviour is observed for other values of $\epsilon^\star$. We note that, differently from eigenstate phase transitions in the context of many-body localization, in the present case the transition occurs at exactly known values of the energy only, and not in a continuous part of the spectrum. This may facilitate future studies, targeting, e.g., exact energy manifolds.

As a last comment, we note that some of the characteristic energies of statistical scars correspond to the energies of the exact scars found in the PXP [29] and in the generalized PXP models [35]. This is not a coincidence: those scars, of the form of matrix product states (MPS), realize an effective "decoupling" of the system in small blocks; the eigenenergies are then originated from the diagonalization of the small blocks, akin to the motifs of statistical scars. Moreover, the participation ratio of exact PXP scars is larger than the typical value of thermal eigenstates (see Appendix G). Despite these similarities, we do not find a direct connection between the two types of scars. In contrast with statistical scars, the number of MPS scars does not grow with the size of the system; moreover, the structure of MPS scars is specific of the low dimensionality of the model. We leave the question of a deeper connection between the two types of scars to future works.

---

[1] For small $N$ we observe that statistical scars can come also from more complicated structures than the ones shown in Fig. 3: for example, we find groups of two or more intersecting motifs. However, for larger sizes, the simple motifs are the most frequent structures.

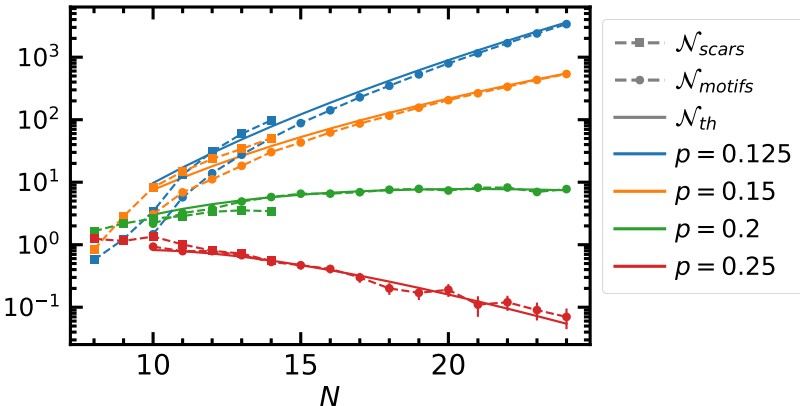

Figure 4: Average degeneracy $\mathcal{N}_{\text{scars}}$ of the eigenspace with $\epsilon^* = 1$, average number of occurrences ($\mathcal{N}_{\text{motifs}}$) of the motif in Fig. 3(a) and expected number of occurrences [$\mathcal{N}_{\text{th}}$, from Eq. (8)] as a function of system size $N$. For $p < p_c \simeq 0.1591$ the degeneracy increases with $N$, while it decays for $p > p_c$.

# 6 Conclusions and outlook

We studied the statistical robustness of a scar manifold by introducing a class of Hamiltonians, Quantum Local Random Networks, that combine locality and constrained dynamics, and that include PXP as a particular instance. Focusing on the giant connected component of a QLRN we have shown that for sufficiently small $p$ it is expected to display *statistical scars*, which occur at special energies $\epsilon^\star = 0, \pm 1, \pm\sqrt{2}, \pm(\sqrt{5}\pm1)/2, \pm\sqrt{3}, \dots$. The latter are solely dictated by random graph theory, and are associated to localized states on certain geometrical motifs on the periphery of the QLRN. A study of the degeneracy of statistical scars for various $p$ as a function of systems size indicates the presence of a quantum phase transition for each special energy $\epsilon^*$ between a phase in which scars proliferate and one in which their number goes to zero for increasing $N$. These states appear in a variety of specific realizations, from (generalized) PXP [29, 35] to Hubbard models [50]. Studying in detail this phenomenon, together with potential generalizations to other QLRN, is an intriguing perspective, that we leave to future investigations.

# Acknowledgements

We thank G. Giudici and J. Goold for discussions, E. Gonzalez Lazo and M. Votto for collaboration on related work, O. Motrunich for comments on the manuscript, and for suggesting the computation of $\mathcal{N}_{\text{th}}$, and A. Lerose for comments on the manuscript, and for pointing Ref. [50] to us.

**Funding information** The work of FS and MD is partly supported by the ERC under grant number 758329 (AGEnTh), by the MIUR Programme FARE (MEPH), and by the European Union's Horizon 2020 research and innovation programme under grant agreement No 817482 (Pasquans).

# A    Generalized Quantum Local Random Networks

The notion of QLRN can be generalized to encompass situations in which either the elementary degrees of freedom are not spin $1/2$ or the number of spins flipped locally is larger than one, as in Ref. [36, 49], maintaining locality and constrained dynamics.

For concreteness, let us consider the set of sequences $\{\sigma_i\}$, where $i = 1, \ldots, N$ and $\sigma_i = \{0, \ldots, q\}$, ($q$ is a positive integer). Two nodes $\{\sigma_i\}$ and $\{\sigma'_i\}$ are connected with probability $0 \leq p \leq 1$ if: *i*) - the string $\{\sigma_i - \sigma'_i\}$ has nonzero entries only locally, i.e. in a compact interval of finite size $L_0 \leq N$ and *ii*) - the distance $\sum_i |\sigma_i - \sigma'_i| \leq S_0$. The random local Hamiltonian associated to this network is then its adjacency matrix and the resulting ensemble of Hamiltonians will be denoted as $\mathcal{H}_p(L_0, S_0)$. Note that if $S_0 \geq 2$ in general the Hamiltonian does not anticommute with the total parity. It is evident that if $q = 1$, $L_0 = 1$ and $S_0 = 1$ we have the special case discussed in the main text and that the PXP Hamiltonian is just one of the realizations in $\mathcal{H}_p(1,1)$. Networks with larger local Hilbert space $q, S_0 > 1$ and more complex spin flips $L_0 > 1$ are naturally related for example to spin-1 models [36] or fermionic models [49], whose analysis is left for future work.

# B    Spectra of QLRN

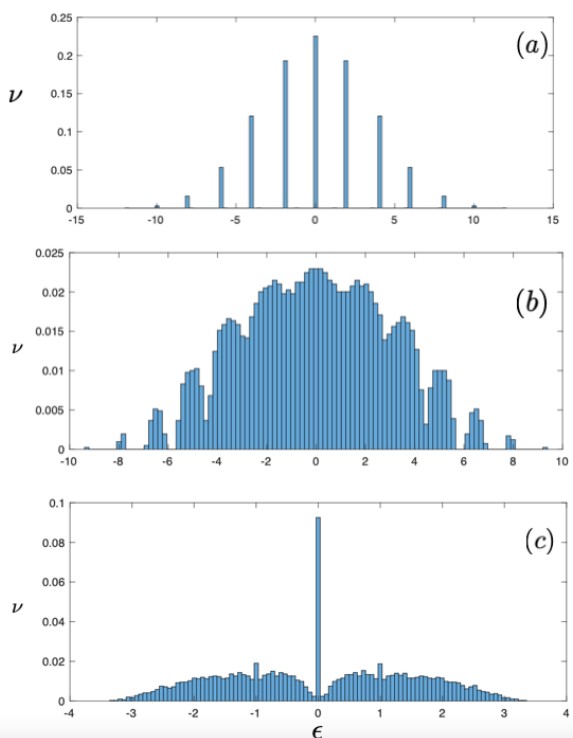

Figure 5: Histograms of the density of states $\nu$ vs. energy $\epsilon$ of the eigenstates for a QLRN with $N = 12$ and $p = 1$ (panel a), $p = 0.75$ (panel b), and $p = 0.15$ (panel c).

Let us now consider the spectra of QLRN as a function of $p$ as shown in Fig. (5) for $N = 12$. When $p = 1$ all states that can be connected by a single spin flip are connected and

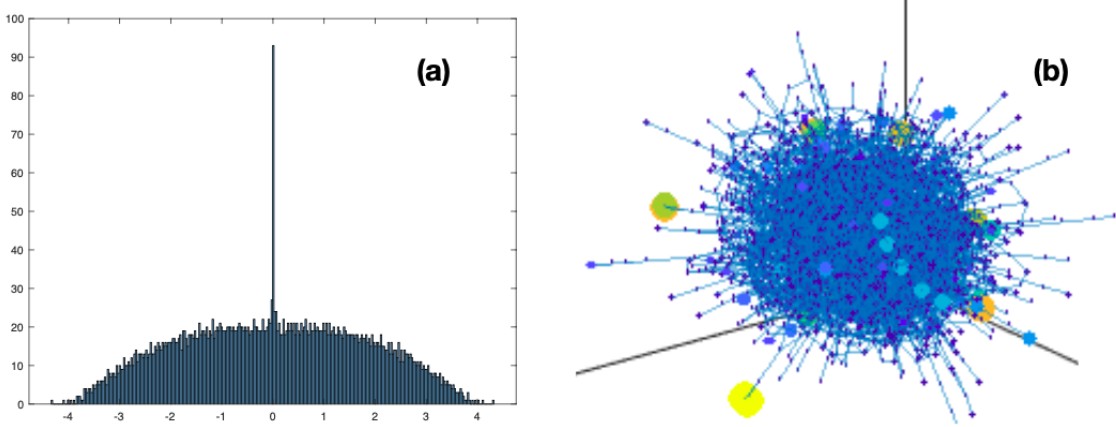

Figure 6: (a) Histogram representation of the spectrum of the giant connected component for a QLRN with $N = 12$ and $p = 0.25$. The peak at $\epsilon = 0$ is still visible and is associated to wavefunctions mainly localized on the periphery of the network as shown in panel (b).

the Hamiltonian is $H = \sum_i \sigma_i^x$: the resulting spectrum is therefore trivial, highly degenerate with eigenvalues $\epsilon_i = N - 2i$, with $i = 0, \ldots, N$, and degeneracy $D_i = \binom{N}{i}$ (see Fig. (5-a)). Introducing a slight stochasticity in the selection of edges splits the degeneracies leading to a characteristic spectrum similar to that shown in Fig. (5-b) for $p = 0.75$. A further reduction of $p$ leads to a fragmentation of the Hilbert space: in the network representation one observes a giant connected component and a few disconnected nodes associated to a peak at $\epsilon = 0$ as well as, for sufficiently small $p$ (Fig. (5-c) for $p = 0.15$), pairs of nodes connected by an edge (peaks at $\pm 1$ in Fig. (5-c) in the histogram of the eigenvalues).

Localization is expected to occur when $p$ is sufficiently small. Of course there is a *trivial* localization related to wave functions completely localized in small disconnected components which will contribute to the peaks at $\epsilon = 0$ and $\epsilon = \pm 1$ in Fig. (5-c). A much more interesting type of localization is however happening in the giant connected component of the network that contains most of the nodes: as shown in Fig. (6) the peak at $\epsilon = 0$ persists also in this case. A visualization of the weights of the corresponding wave functions in the network, shows that these localized states are associated to wave functions with large amplitudes on nodes at the boundaries of the network. Qualitatively similar results are obtained for different $N$.

## C   Hilbert space fragmentation

To obtain the connected components of the graph of the QLRN we use the Python package *networkx*. In the range of probabilities $p$ that we consider, our numerical analysis (see Figure 7) is compatible with the scenario of weak fragmentation [36, 51]: the average number of connected components grows as $a^N$ with $a < 2$, while the ratio between the dimension of the largest connected component and the total Hilbert space dimension approaches 1 in the thermodynamic limit. For the purpose of this work, we will focus solely on the spectrum of

the largest connected component.

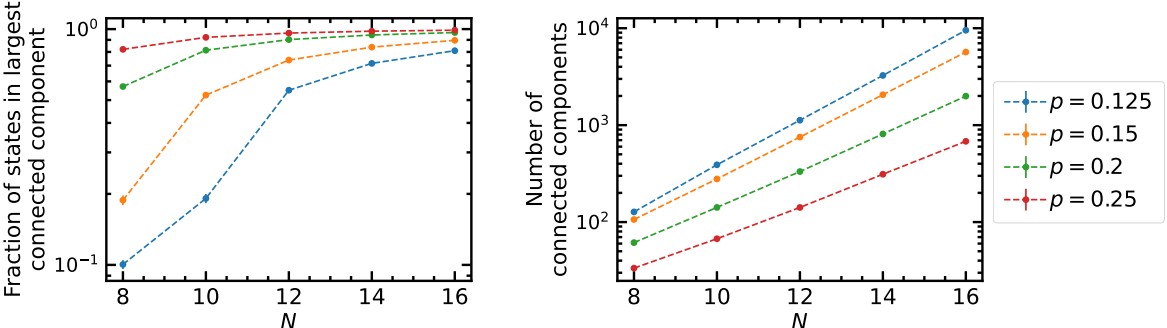

Figure 7: Left: The fraction of states in the largest connected components goes to 1 for large $N$. Right: The number of connected components grows as $a^N$, with $a < 2$.

## D   Participation ratio and system size

In Fig. (8) we plot the participation ratio of the eigenstates for different values of the system size $N$. We note that, as $N$ is increased, the majority of the eigenstates get closer to a smooth dependence of $\mathcal{P}$ on the energy $\epsilon$ (the thermal cloud). Statistical scars, instead, remain well isolated, with strongly non-thermal values.

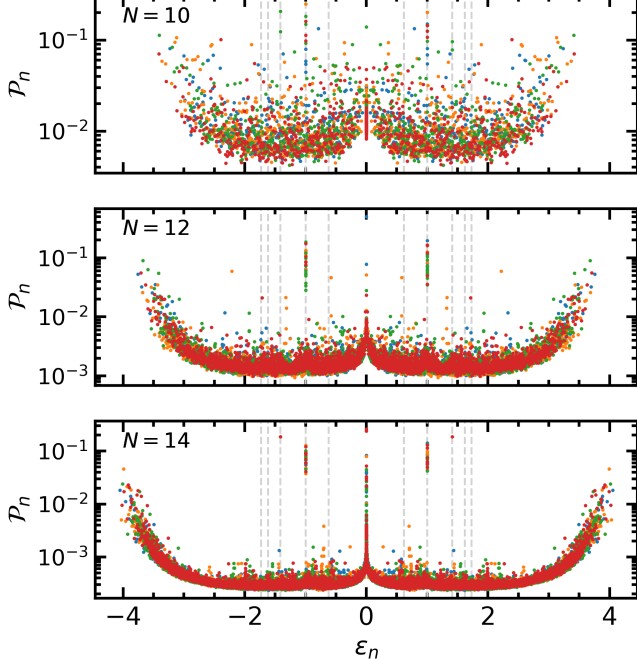

Figure 8: Participation ratio $\mathcal{P}_n$ of the eigenstates for $p = 0.2$ and different system sizes $N$.

## E   Centrality and degree of statistical scars

The characterization of the localization of stochastic and statistical scars done in the main text with the participation ratio $\mathcal{P}_n$ and the betweenness $B_n$ can be done using other figures of merit such as the degree and the centrality of the eigenstates, defined as

$$\langle k \rangle_n = \sum_i |c_n(\{\sigma\})|^2 k(\{\sigma\}), \tag{6}$$

$$\langle C \rangle_n = \sum_i |c_n(\{\sigma\})|^2 C(\{\sigma\}). \tag{7}$$

As shown in Fig. (9), statistical scars are characterized by anomalously small values of both quantities.

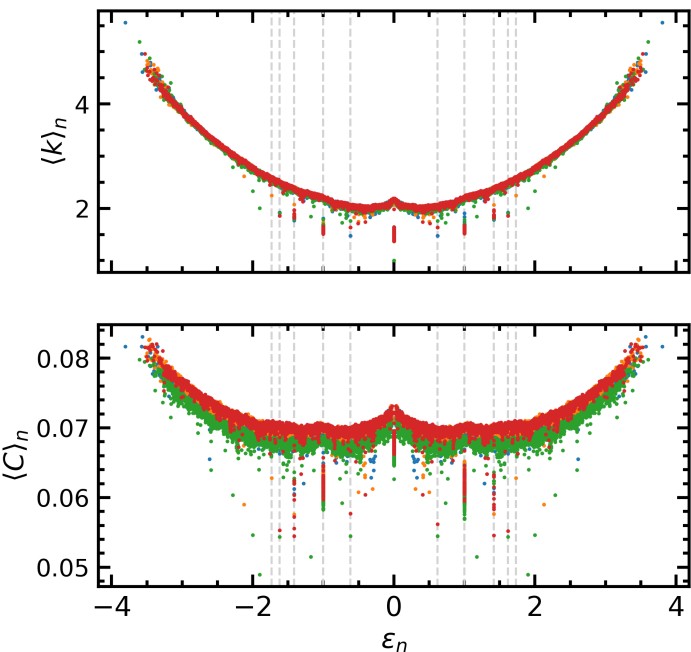

Figure 9: Degree $\langle k \rangle_n$ and centrality $\langle C \rangle_n$ of the eigenstates as a function of their energy $\epsilon_n$. Different colors refer to different realizations of the network. Statistical scars are chacterized by small values of both $\langle k \rangle_n$ and $\langle C \rangle_n$.

## F   Eigenstate phase transitions

In the main text, we discussed for the presence of an eigenstate phase transition based on the degeneracy of statistical scars at $\epsilon^\star = 1$. It is possible to extend this picture to all network-predicted values of quantized energies.

In Fig. 10, we show the degeneracy scaling ($\mathcal{N}_{\text{scars}}$, marked by squares) versus system size for three additional values of $\epsilon^\star$. For small $p$ we consistently observe that degeneracy is

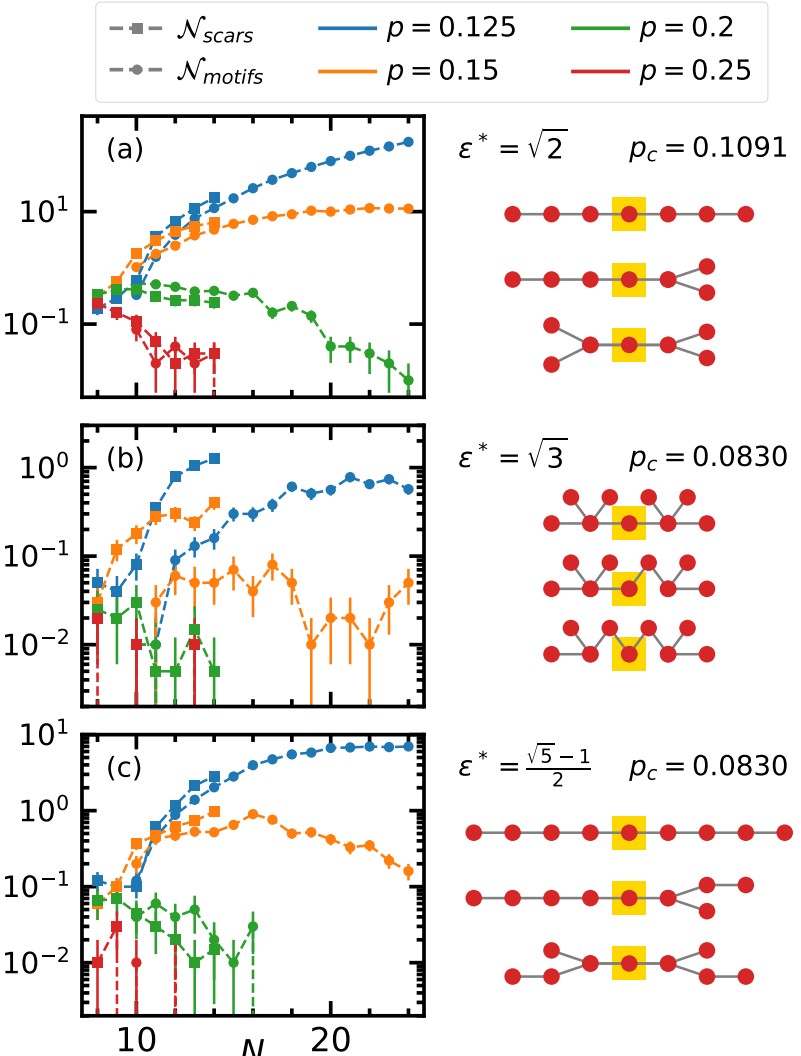

Figure 10: Degeneracy of statistical scars (squares) and occurrences of motifs $\mathcal{N}_{\mathrm{motifs}}$ (circles) vs. system size $N$ for different $p$ for 100 realizations of the QLRN. The motifs that are counted for each eigenvalue $\epsilon^*$ are shown in the right panel. The "root node" (marked by a yellow square) is the only one in the motif that is connected with the rest of the network.

increasing with system size. To better analyze the critical value of $p$ for the different energies, we count the occurrences of the motifs associated with statistical scars ($\mathcal{N}_{\mathrm{motifs}}$, marked by circles in Fig. 10): the number of these motifs represents a lower bound on the number of scars; we expect that, close to the transition, the number of scars coincides with the number of motifs in the thermodynamic limit. We observe that – with the exception of the case $\epsilon^* = \sqrt{3}$ – the number of occurrences of the motifs is in good agreement with the total degeneracy of the scars, confirming our expectation that statistical scars are associated with the presence of these motifs.

From this observation, we can give an analytical estimate of the transition by counting

the expected number of occurrences of a certain motif. Let us first consider the single motif associated with $\epsilon^* = 1$ in Fig. 3-(a): the expected number of occurrences has the form

$$\mathcal{N}_{th}(\epsilon^* = 1) = 2^N (1-p)^{4N-6} p^4 \frac{N(N-1)}{2}(N-1)^2 \tag{8}$$

where the the factor $2^N$ counts the possible choices of the "root" node $C$, the factor $p^4$ comes from the 4 edges, the factor $(1-p)^{4N-6}$ is the probability that every other edge that can come out of the nodes $A, B, D, E$ is absent, and the last terms are combinatorial factors that count the possible choices of the nodes $B, D$, and $A, E$. As shown in Fig. 4, The scaling of Eq. (8) is in perfect agreement with the numerics. From Eq. (8) we find that the transition occurs at $p_c = 1 - 2^{-1/4} \simeq 0.1591$. To generalize this argument to the other values of $\epsilon^*$, we note that the number of occurrences is in general proportional to $\mathcal{N}_{th} \propto 2^N (1-p)^{mN} N^m$ where $m$ is the number of nodes (excluding the root node, marked by a yellow square in the right panels of Fig. 10) in the motifs and terms subleading in $N$ are neglected. We hence obtain the transition probability $p_c = 1 - 2^{-1/m}$. As can be seen in Fig. 10, the number of occurrences of the motifs increases also for values of $p > p_c$ for the accessible system sizes. This happens because, for values of $p$ not too far from $p_c$, the scaling of $\mathcal{N}_{th}$ may be dominated by the power-law term at these system sizes. In fact, for $p > p_c$, $\mathcal{N}_{th}$ has a maximum at system size

$$N_{\max} = - \left[ \log\left( \frac{1-p}{1-p_c} \right) \right]^{-1}. \tag{9}$$

Since our largest system size is $N = 24$, we expect to observe a decaying trend only for $p > 1 - (1-p_c)^{-1/24}$, i.e., $p > 0.193$ for $\epsilon^* = 1$, $p > 0.145$ for $\epsilon^* = \sqrt{2}$, $p > 0.120$ for $\epsilon^* = \sqrt{3}$ and $\epsilon^* = (\sqrt{5} \pm 1)/2$. For all these values, we observe perfect agreement with our numerical data in Fig. 10.

## G    Participation ratio in the PXP model

We now compute the participation ratio of the exact scars reported in Ref. [29] and compare it with the value of thermal eigenstates.

For an unnormalized state $|\psi\rangle$ the participation ratio can be computed as

$$\mathcal{P}_{|\psi\rangle} = \frac{I_4(|\psi\rangle)}{[I_2(|\psi\rangle)]^2}, \tag{10}$$

where we defined

$$I_q(|\psi\rangle) = \sum_{\{\sigma\}} |\langle\{\sigma\}|\psi\rangle|^q. \tag{11}$$

Using the same notation as in Ref. [29] to label the scars, we find

$$I_q(|\Phi_1\rangle) = I_q(|\Phi_2\rangle) = (2^{q/2}+1)^{L_b} + (2^{q/2}-1)^{L_b} + (2^q - 1)[1 + (-1)^{L_b}], \tag{12}$$

$$I_q(|\Gamma_{\alpha\beta}\rangle) = (2^{q/2}+1)^{L_b} - 2 + [1-(-1)^{\alpha+\beta+L_b}]^q, \tag{13}$$

for $L_b = L/2$ and $L$ (even) is the system size. We obtain that, in the limit of large $L$, the participation ratio decays as $\mathcal{P}_{\text{scars}} \propto (\sqrt{5}/3)^L \sim (0.745)^L$ for all the exact scars, while it

decays as $\mathcal{P}_{\text{random}} \propto \mathcal{D}_L^{-1} \sim \phi^{-L} \sim (0.618)^L$ for a random state in a Hilbert space of the same dimension $\mathcal{D}_L$. This proves that the participation ratio of scars decays much slower than the one of random states. Moreover, as shown in Fig. 11, the participation ratio of the exact scars in the PXP model is significantly larger than the typical value of thermal eigenstates.

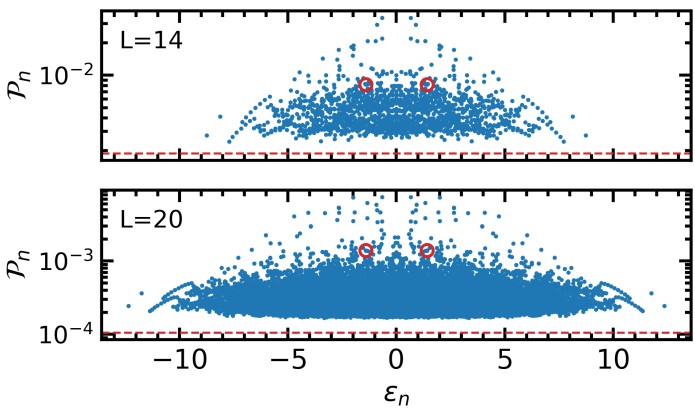

Figure 11: Participation ratio of the eigenstates in the PXP model with $L$ sites and open boundary conditions. Red circles indicate the exact scars $|\Gamma_{12}\rangle$, $|\Gamma_{21}\rangle$ with energies $\pm\sqrt{2}$. The dashed red line indicates the participation ratio of a completely random state in a Hilbert space with same dimension.

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
