# Peer review of "Quantum local random networks and the statistical robustness of quantum scars"

_SciPost Physics_

## Round 3 · Referee Report · Anonymous · 2023-1-22

Report

I thank the authors for modifying the manuscript and for replying my comments and questions. I believe the authors have addressed my questions and the readability of the paper indeed has improved some in this modified version.

I therefore recommend the publication of this work on Scipost Physics, though I would still suggest the authors to spend a bit more effort on giving the readers more illustrations and (toy) examples of the “degree”, “centrality” and “betweenness”, helping the physicist readers to build up physical picture and meaning of those graph-theoretical measure.

---

## Round 3 · Author Response

Reply to Referee 1: We thank the Referee for their careful reading and positive assessment of the manuscript, and for their constructive comments on the physical content and the presentation. Here we address their specific questions and comment:

  1. That is correct: most of the quantities we compute depend on the particular choice of superposition. We have not disentangled the degenerate states obtained from ED, so the eigenstates we found have support, in general, on more nodes than the simple motifs. In light of this, Figure 1c and 1d give some "global" information about the states in the degenerate subspace, for example the fact that they tend to localize in the periphery of the network, while the values of single states in the subspace are not immediately informative. It would be interesting to find a systematic way to disentangle those states, but we could not find a simple procedure that could be applied efficiently at large system size. Therefore, rather than constructing the most localized superpositions with a general procedure, we checked their presence "a posteriori" by counting the motifs in the graphs: for each motif, a localized eigenstate exists as an appropriate superposition in the degenerate eigenspace; all the properties of the corresponding eigenstate (degree, centrality, betweenness, etc.) can then be easily inferred from the structure of the motif.

  2. We thank the Referee for their comment: indeed, the presence of the delta peak in the density of states is not sufficient to prove localization. To claim that the origin of the peak is associated with localized states we then consider other quantities, such as the participation ratio. Nevertheless, we find it suggestive that a zero energy delta peak occurs in other random networks (Ref. 44), where similar localized states also emerge. We rephrased the text to clarify this point. We also noticed that our observation about the origin of the exponential degeneracy at $\epsilon=0$ as a consequence of the theorem in Ref. 41 was incorrect: QLRNs are in general not invariant under inversion, so the theorem does not apply.

  3. To obtain the connected components of the graph we use the Python package networkx. In the range of probabilities p that we consider, our numerical analysis (see figure attached/Figure 7 in the new draft) is compatible with the scenario of weak fragmentation [PRX 10, 011047 (2020); PRX 12, 011050 (2022)]: the average number of connected components grows as a^N with a<2, while the ratio between the dimension of the largest connected component and the total Hilbert space dimension approaches 1 in the thermodynamic limit.

  4. To count the number of motifs we used the following algorithm. We first identified all the nodes with degree 1 (this was done with the package networkx in Python). For each of these nodes, we check the degree of the node connected to it: if it is 2, 3, or 4, we further check the other nodes connected to it. Iterating this procedure with slight variations depending on the specific type of motifs that we want to count, we can identify all the submotifs that are connected to the network through a single node. We then count the motifs by counting the pairs of submotifs that share the same "root node" (in yellow in Fig. 3 and 10).

  5. We thank the Referee for their comment. We have now added the definition of "degree" and expanded the discussion on the properties of the nodes.

6-8. We thank the Referee for their suggested changes, we amended the text accordingly.

Reply to Referee 2: We thank the Referee for reading the manuscript and for their constructive criticism. We agree that the presentation was not clear enough. We nevertheless believe that a short summary of the main properties of the PXP model and its quantum scars is necessary to motivate our definition of Quantum Local Random Networks (QLRNs). To improve the clarity, we split Section 2 into two different sections: in the first one, we briefly introduce the PXP model and its quantum many-body scars, in the second one we introduce the QLRNs, that are the main subject of our investigation. We also included an explicit Hamiltonian for the QLRNs. To further improve the clarity of the manuscript, we now moved the discussion on the motifs from the caption to the main text, and we included a sentence to explain the meaning of betweenness centrality.

As the Referee correctly emphasises, the Hamiltonians of QLRNs do not in general have a representation as a sum of terms with finite support. As we specify in the text, what we mean by ``local" is that the Hamiltonian has non-zero matrix elements only between sites that differ by a single spin flip: such locality is reflected in the specific form of the network space, and not referred to microscopic dynamics (which, indeed, generically requires projector operators that span the entire system). This guarantees the extensivity of the Hamiltonian. We hope the text is now unambiguous.

For what concerns the relevance to the PXP model, we remark that, while QLRNs do not share the same representation as simple (i.e., with few-body terms) many-body Hamiltonian as the PXP model, they are very similar to the PXP when seen as networks: the PXP is, in fact, a subgraph of the N-dimensional hypercube graph, and QLRNs are defined as an ensemble of subgraphs of the N-dimensional hypercube graph.

The results of our work shed some light on the role of constraints (understood as the suppression of the connectivity of the network) as the origin of quantum scars. The fact that statistical scars share some features with PXP scars is not our fundamental point, but suggests that further work in this direction could lead also to a deeper understanding of the PXP model.

One of the lessons we learn from QLRNs is that by reducing the connectivity of the network (i.e., enhancing the constraints) it is possible to obtain a transition from a regime without scars to an exponential number of them. We are not aware of connections with anything well-known in the literature, other than the references already included in the manuscript. We would be grateful if the Referee could provide some references for the mentioned connections.

Reply to the requested changes 1. We thank the Referee for the suggestions. We amended the text accordingly. 2. We now included a sentence to specify what we do and do not mean for ``local". 3. We are not aware of any connection with SAT-UNSAT transition. We would be grateful to the Referee if they could be more specific and explain the intuition for this possible connection. 4. We decided to maintain a discussion on the PXP model in the first section after the introduction, as it serves as a motivation for our definition of QLRNs. The discussion is now in a separate section from the definition of the QLRNs.

---

## Round 3 · List of Changes

1. We split Section 2 into two different sections: in the first one, we briefly introduce the PXP model and its quantum many-body scars, in the second one we introduce the QLRNs.
2. We included an explicit Hamiltonian for the QLRNs.
3. We added references to the Appendices.
4. We rewrote the discussion on the zero energy delta peak.
5. We added the definition of "degree" and expanded the discussion on the properties of the nodes.
6. We moved the discussion on the motifs from the caption of Fig. 3 to the main text.
7. We corrected the typo in the caption of Fig. 1.
8. We added a section in the appendix (Appendix C) on Hilbert space fragmentation.

---

## Editorial Decision

in_voting